# Knowledge and Acceptance of the COVID-19 Vaccine for COVID-19 Disease Prevention among the Indian Population: A Mixed-Method Study

**DOI:** 10.3390/vaccines10101605

**Published:** 2022-09-24

**Authors:** Pratibha Taneja, Charu Mohan Marya, Parul Kashyap, Sakshi Kataria, Ruchi Nagpal, Mohmed Isaqali Karobari, Anand Marya

**Affiliations:** 1Department of Public Health Dentistry, Sudha Rustagi College of Dental Sciences and Research, Faridabad 121001, Haryana, India; 2Department of Restorative Dentistry and Endodontics, University of Puthisastra, Phnom Penh 12211, Cambodia; 3Department of Conservative Dentistry & Endodontics, Saveetha Dental College & Hospitals, Saveetha Institute of Medical and Technical Sciences University, Chennai 600077, Tamil Nadu, India; 4Department of Orthodontics, Faculty of Dentistry, University of Puthisastra, Phnom Penh 12211, Cambodia; 5Department of Orthodontics, Faculty of Dental Medicine, Universitas Airlangga, Surabaya 60115, Indonesia

**Keywords:** knowledge, acceptance, COVID-19, transmission, prevention

## Abstract

Aim: To assess the Knowledge and Acceptance of the COVID vaccine among the Indian population. Materials and methods: The present mixed-method study was conducted in two phases. The first phase: quantitative assessment of knowledge and acceptance for the COVID-19 vaccine using an E survey (N = 606). The second phase: qualitative assessment using semi-structured face-to-face interviews with the study participants (N = 30) and assessment was done using a thematic approach. Study participants were selected using the convenience sampling method. Results: It was found that a large proportion of subjects in the 16–25 year of age group knew the cause of disease. But knowledge about its transmission process was found to be more in >60 years of age gap and almost all the participants in all the age group preferred Covishield. The vaccine acceptance rate was found to be low as compared to the knowledge. Conclusion: Most study participants were found to have satisfactory knowledge, but acceptance rate was comparatively lesser. Hence, more information and awareness campaigns must be launched reassuring the population about vaccine safety.

## 1. Introduction

The coronavirus disease COVID-19 emerged in Wuhan, China at the end of 2019. Since then, it has spread to 200 countries and has been declared a global pandemic by the World Health Organization (WHO) [1]. Globally more than 284,907,960 positive COVID-19 cases were recorded with at least, 5,438,615 deaths. In India more than 3.48 Cr positive COVID-19 cases were recorded, Total Deaths of 4.81 L people with a total no of recovered cases of 34,251,292 till December 30, 2021 are recorded [1].

The second wave of COVID-19 in India brought unprecedented losses. The poorest and the most marginalized, faced more risks without the means to absorb the economic shocks and mitigate the health crisis [2]. Despite the prospect of a third wave, they are caring for their families, maintaining their livelihoods, and leading efforts to combat the pandemic [3]. As rapid human-to-human transmission occurred and much about the virus remained unknown, lockdown precautions were deemed required to halt the pathogen’s spread. Because of the virus’s obscurity, there has been a lot of misinformation and misunderstanding regarding the virus, how it spreads, and the steps that should be taken to avoid infection [4]. This becomes increasingly challenging with the vast amount of misinformation and disinformation shared on social media that is clouding people’s understanding of COVID-19 [5].

COVID-19 vaccinations have already reached billions of individuals around the world, and the evidence is overwhelming that they provide life-saving protection against a disease that has killed millions [6]. The pandemic is far from over, and they are our best bet of staying safe. Till date total dose given are 83.7 Cr, people who completed their first dose are 60.6 Cr and 23.1 Cr. are fully vaccinated. But, in India, we have a very huge dense population without well-established medical facilities, which is a matter of concern. The government’s concern during the lockdown is that a large percentage of people are illiterate, isolated, migrants, live remotely, and are below the poverty line, trying to meet their necessities.

COVID-19 beliefs are based on a variety of sources, including preconceptions about other viral infections, government information, social media, and the internet, prior personal experiences, and medical sources. The correctness of these ideas may influence different preventative behaviors and may vary across the community. In many circumstances, a lack of understanding, or the fact that most medical beliefs are misinterpreted or untrue, might pose a risk. With the widespread availability of smartphones, more individuals in LMICs can now access the internet and social media [7]. Although this can be a useful tool for self-education, which is an important part of vaccination decision-making, it also comes with a number of drawbacks, including misinformation and incomplete data, as well as inconsistent and complicated scientific information that can be difficult to understand [8]. The causes behind COVID-19 vaccination acceptance and skepticism are still unknown. Continued research on COVID-19 vaccine acceptance and hesitation should be a priority in the battle against this pandemic because the globe shares a joint responsibility in battling this pandemic. This information should then be used to support contextualized advertising and information exchange, resulting in enhanced trust in and uptake of accessible vaccines.

The COVID-19 pandemic is sweeping the globe. It’s been two years since the first COVID-19 case was discovered. Much has evolved in terms of knowledge about the disease and its treatment during this time [9]. Omicron, an unique strain first discovered in Botswana, sent Europe onto high alert after instances were discovered in the United Kingdom, Germany, Italy, and Belgium. In India, the coronavirus Omicron variant has also been detected [10,11]. Despite specialists highlighting the necessity of vaccines, it was initially reported in India by Karnataka’s health minister, who noted that a sample from one of the two recent South African returnees appeared to be “different from the Delta form”. According to data updated by the Union health ministry, a total of 1892 cases of the Omicron form of coronavirus have been discovered across 23 states and Union Territories, with 766 of them recovering or moving. The state with the highest cases (568) is Maharashtra, which is followed by Delhi (382), Kerala (185), and Rajasthan. Mumbai Mayor Kishori Pednekar stated on Tuesday that if the city’s daily COVID-19 cases reach 20,000, a lockdown will be implemented in accordance with Union government regulations. Pednekar told reporters in her office at the Brihanmumbai Municipal Corporation (BMC) headquarters that citizens should wear triple-layer masks on public buses and local trains.

Getting COVID-19 vaccination is the only way by which we can try our best to protect ourselves and others from this deadly disease [12]. A possible barrier to this could be vaccine hesitancy despite the availability of vaccination services. Vaccine hesitancy, defined as “a delay in accepting or refusing immunization despite the availability of vaccination services,” could stymie COVID-19 vaccination attempts in the future [13]. This might be due to the that people who are not accepting COVID-19 vaccine have a lack of knowledge about vaccine, unavailability of vaccine slots and center and there are also so many rumors and myths about the vaccination few of them are: In women, some people thought The COVID-19 vaccine causes infertility in women, they have already been diagnosed with COVID-19, they can get COVID-19 from the vaccine so they don’t need to receive the vaccine, they are not at risk for severe complications of COVID-19 so they do not need the vaccine, If they receive the COVID-19 vaccine, they are at a greater risk to become sick from another illness, Certain blood types have less severe COVID-19 infections, so getting a vaccine isn’t necessary. And there are also rumors that the COVID-19 vaccine is not safe in pregnant or lactating women, children below 18 years, and aged individuals with some systemic illness. A review by Silver LE et al., reported to vaccine adverse events reporting system from 1990 to 1997 found that almost half were attributable to sudden infant death syndrome (SIDS), which decreased in frequency following recommendations in the early 1990s to change infant sleep environment yet no data regarding COVID 19 vaccination is linked to SIDS till date [14]. Also a study by Passarelli-Araujo et al. found that the fatality risk experienced by an unvaccinated person is 7.1-fold higher than in a fully vaccinated subject. Hence vaccination becomes one of the most feasible, approachable and accessible way to protect ourselves from vaccine preventable diseases [15].

The COVID-19 vaccination is the only option to protect ourselves and others from this COVID-19 disease. The success of any vaccination program to achieve herd immunity is determined by vaccine knowledge, acceptance, and uptake rate, which is why the current study was carried out with the goal of determining “Knowledge and Acceptance of COVID-19 vaccine for COVID-19 disease prevention among the Indian population”.

## 2. Materials and Methods

Ethical clearance was sought from one of the reputed dental colleges of NCR, Delhi. After presenting the aim, objectives, and procedures of the study. The aim and objective of the study were explained, and informed consent was obtained from the participants before the commencement of the study.

### 2.1. Sampling and Recruitment

The present mixed-method study was conducted in two phases. The first phase was composed of a quantitative assessment of knowledge and acceptance for the COVID-19 vaccine using an E survey. All the questions in the survey link were presented in English as well as in the Hindi language to enhance comprehensibility. Content validation was done by two field experts. A pilot run was done among 10 participants to assess the validity and feasibility of the questionnaire.

The second phase consisted if a qualitative assessment using semi-structured face-to-face interviews with the study participants. Study participants were selected using the convenience sampling method coming to the Outpatient department of one of the reputed Dental colleges of North India.

### 2.2. Quantitative Phase

The study was conducted among subjects with more than 18 years of age as vaccination till 6 October 2021, was not allowed for minors. The sample was procured from the outpatient department of one of the reputed Dental colleges of North India using convenience sampling from 3 July 2021 to 6 October 2021. Participants were provided an information sheet explaining the aim, objectives, and purpose of the present research. Participants who agreed to participate in the research were provided an E survey link there itself and were asked and guided accordingly to fill up the survey. Link of E survey was circulated through Instagram, WhatsApp, text message, and phone. The survey included 12 questions in total to assess the knowledge and acceptance of the Indian population towards the COVID-19 vaccine. Sociodemographic details of the participants were also captured.

### 2.3. Qualitative Phase

After the completion of quantitative data collection, the second phase of this study was initiated. The principal investigator was trained by a professional in the field to execute qualitative interviews. The topic of the interview was predefined whereas, the sequence of the questions was kept flexible based on the patient’s response. A flexible approach provides the interviewers the opportunity to probe deeper into certain topics of interest prompted by the participants. For e.g., if a participant reported that one of his family members was affected by this COVID-19 pandemic, he/she can be further asked about the impact of disease, vaccination status, and then participants were purposely selected based on their survey responses to obtain a representative sample so that findings can be generalized to all Indians. The required information was collected using a semi-structured pattern. Main points in the interview guide to various questions regarding their knowledge regarding the beneficial and adverse effects of COVID-19 vaccination and regarding acceptance and barriers faced while getting vaccinated.

### 2.4. Interview Procedure

A face-to-face interview was conducted with selected participants of the survey. Interviews were carried out in the out-patient department only which lasted for up to 5 to 10 min. All the interviews were taken by the primary investigator and were audio-recorded and translated to the English language.

### 2.5. Data Management and Analysis

Quantitative data analysis was performed using Statistical package for social science software version SPSS-21 IBM Inc., Armonk, NY, USA. Chi-square was used for gender-wise comparison of knowledge and acceptance for COVID-19 vaccination. Binary logistic regressions were performed when the dependent variable had two groups. For each regression, the unadjusted odds ratio (OR) and adjusted odds ratio (AOR) with their 95% confidence interval were presented. For qualitative analysis, face-to-face interviews of 30 volunteers were transcribed in full. Face-to-face interviews were taken until redundant saturation was attained. We did not use any software to code our data. Transcripts were manually analyzed and coded. A code of the list of major themes was developed manually using the deductive approach for content analysis & was then compared and cross-checked with additional interviews. A final agreed theme was applied to all interviews.

## 3. Results

### 3.1. Age Group Wise Distribution of Study Participants Based on Their Knowledge Regarding COVID-19 Disease

Variation in the responses were observed while analyzing the results. Though larger proportion of subject in 16–25 year of age gap knew the cause of disease. But knowledge about its transmission process was found to be more in >60 year of age gap and almost all the participants in all the age group preferred Covishield. (Table 1)

### 3.2. Gender Wise Distribution of Study Participants Based on Their Knowledge Regarding COVID-19 Disease

Gender wise comparison showed quantitatively more knowledge about COVID-19 disease, its etiology, transmission process and preventive measure among males as compared to females. (Table 2)

### 3.3. Age Group Wise Distribution of Study Participants Based on Their Acceptance for COVID-19 Vaccine

Despite the deadly nature of the disease, approximately 10% of the study participants were not vaccinated. Almost all the study participants believe that even after getting vaccination they could still get COVID disease.

Almost one third of the population was afraid of vaccination because of its side effects. Associated comorbidities were also found to be one of the barriers against COVID- 19 vaccination. Study population was found to be skeptical regarding vaccination safety among pregnant females, lactating women, and children under 18 years of age. (Table 3)

### 3.4. Gender Wise Distribution of Study Participants Based on Their Acceptance for COVID-19 Vaccine

Gender wise comparison showed quantitatively more acceptance about COVID-19 disease, vaccination among males as compared to females (Table 4).

### 3.5. Overall Knowledge

Overall knowledge of the study participants was found to be sufficient regarding COVID-19 vaccine for COVID- 19 disease prevention among Indian population. (Table 5). Despite having ample knowledge many of the participants were found to be skeptical for vaccination probably due to the side effects occurring post vaccination. (Table 6).

### 3.6. Qualitative Phase

After completion of the first phase of the study i.e., complete quantitative data extraction, the second phase of the study was carried out. In this phase qualitative interviews were carried out among 30 participants till redundant saturation was achieved.

The qualitative phase is summarized in 2 major domains; 1. Knowledge: a. Transmission process and b. Prevention and 2. Acceptance: a. No acceptance, b. Partial acceptance and c. Complete acceptance.

### 3.7. Knowledge of COVID-19 and Its Transmission

P1. “I knew COVID-19 is spreading speedily. All I know is I don’t want to get affected by this disease. That is why I am taking all the precautions like social distancing, hand hygiene, masking, gloving, vaccination needed to avoid this deadly virus”.

P2. “My family members and I did not step out for 3 months as we all were very afraid of this deadly disease still, we got attacked by this. So, as per my experience vaccination is the only way to protect ourselves from this. Does not matter which vaccine is available. All are proven efficacious. All that matters is you should be vaccinated”.


Prevention


Prevention allows people to understand repercussions, helps make informed and healthy choices in this present study 95.3% of the study participants know the preventive measures to prevent COVID-19 disease, and 57.1% prefer Covishield vaccine for vaccination.

P3. “I do not remember which vaccine slot was given to me. But all I remember is I was vaccinated as soon as it was made available for us by the government. And by God’s grace and vaccination, I never got COVID-19 disease”.


Acceptance assessment of COVID-19 vaccine


83.4% of the study participants were vaccinated but only a few of them were know about the side effects of the COVID-19 vaccine and out of 606 participants, 65% were afraid of getting vaccination all because of the side effects of the COVID-19 vaccine.


No acceptance (safety concern)


P4. “I was a little anxious about getting vaccinated during the initial phase after its launch. As I was suffering from diabetes and hypertension. Despite wanting to be vaccinated I could not but later, after the government gave approval, I got myself vaccinated”.

P5. “I have seen my friends saved from COVID-19 pandemic because they were vaccinated. But, I have also seen them getting adverse symptoms post-vaccination which makes me anxious to get vaccinated”.

P6. “I got vaccinated as soon as government made it available for all, but I could not get my wife vaccinated as she was pregnant later she suffer from COVID-19, in this pandemic I lost my wife as well as my unborn child”.


Complete acceptance


P7. “The efficacy of the vaccine has been proven so far by so many trials conducted by the government and also, I have seen people not getting attacked by this deadly virus as they were vaccinated. So, only as soon as the government opened the vaccination program for the general public. I got myself vaccinated.”

P8. “I am myself vaccinated and recommend all human beings above 18 years to get the vaccine if they want to survive. During the peak, COVID-19 pandemic attack, all of my family members contracted COVID disease except me. Till then, I had taken only my first dose of COVID vaccination yet I was saved from this deadly virus”.


Barriers


Socio-cultural, religious, and economic related barriers were identified for the practice of COVID-19 prevention measures. The present study findings suggest the need to strengthen community awareness and education programs about the prevention measures of COVID-19 and increase diagnostic facilities with strong community-based surveillance to control the transmission of the pandemic.

P9. “I want to get vaccinated but whenever I try to book a slot on Covin/Arogya or Umang app but I did not get any slot”.

Collaborating with quantitative findings, qualitative findings project good knowledge about COVID- 19 vaccine for COVID- 19 disease prevention among the Indian population.

## 4. Discussion

In India, there are currently 13 lakh positive cases. In the fight against the COVID-19 epidemic, the entire world bears a common duty [16]. And, as we all know, the second wave of COVID caused everyone to suffer unprecedented losses. The only way to protect ourselves and others from this fatal disease is to get the COVID-19 vaccine. The success of any vaccination program to achieve herd immunity depends on the vaccine knowledge, acceptance, and uptake rate that is why the present study was conducted with the aim to assess the “Knowledge and Acceptance of COVID-19 vaccine for COVID-19 disease prevention among the Indian population”.

The issue of vaccination hesitancy must be addressed to develop herd immunity. The Indian government has also scaled up the vaccine promotional programs, particularly through social media and mass media [17]. That is why we also choose social media platform for the first phase of the study in a hope of a high response rate for COVID-19.

Two consecutive studies were conducted to assess the knowledge and acceptance of the COVID-19 vaccine among the general population. The quantitative phase of the study yielded a response rate of 89.1%. Respondents of the current study believed that they were knowledgeable about the importance of being vaccinated and about the side effects that may occur post-vaccination. Similar results were reported by Cordina et al. [18] in a study conducted to assess the attitude towards COVID-19 vaccination, vaccine hesitancy, intention to take the vaccine. In their study, the authors discovered a strong link between participants’ desire to receive the vaccine and their belief that vaccination will protect them from COVID-19.

The availability of increased information regarding vaccine efficacy and safety is rivalled by the misinformation spreading on social media [18]. The WHO is doing its utmost to identify and address the misinformation, but it is ultimately up to individuals to not believe false information [19]. Also, gender has been demonstrated to be a significant issue during this pandemic. The results of the present study showed that a greater number of females knew about the transmission process of COVID-19 disease and had lesser fear to get vaccinated because of COVID-19 vaccination side effects. Contract to this Cordina et al., Wang et al. and Murphy et al., Fisher et al. and Khunbchandani et al. found that more females were unsure of taking the vaccination [18,19,20,21,22].

Age also presented to be a significant barrier in the vaccination process. Less number of Subjects more than 60 years of age were found to be vaccinated in the present study. This could be attributed to the fact that the composed subjects among this age group were already less, or chronic comorbidities stopped them to get vaccinated. Contrary to this Cordina et al., Murphy et al. and Szilagyi et al. in their study concluded that respondents between the ages of 30–49 appeared to be less willing to take up the vaccine [18,20,23]. Murphy also stated that the information about the elderly’s vulnerability to COVID-19 was apparent and that it had been reported [18].

The qualitative assessment also corroborated with the quantitative assessments i.e., most of the study participants were having knowledge of COVID-19 disease, its transmission processes, and its preventive measures yet attitude towards vaccination was found to be compromised because of safety concerns, socio-cultural, religious, economic and slot availability for a free vaccination. Despite a thorough literature search, we could not find any study demonstrating the qualitative assessment of acceptance and knowledge regarding COVID-19 vaccination. Hence the results of this phase of the study could not be compared with other studies.

While a copious amount of information is available on social media platforms a lot may be misleading. Thus, increased responsibility of national or international health care organizations as well as of oneself is important to take care of fraudulent information hence globally effective campaigns should be run to make everyone know about the need for COVID-19 vaccination against this pandemic [24,25,26].

The mixed-methods nature of the study gave a new horizon to captivate the participant’s responses who could not actually quantify their knowledge and acceptance towards vaccination. The qualitative approach of conducting interviews provided a factsheet of what is stopping them from getting a vaccination. No research comes without limitation. One of the major limitations of the present research is sample was recruited through online social media platforms Those in the lower or lower-middle classes who need to be vaccinated lack understanding of social media platform use and the importance of being vaccinated.

## 5. Conclusions

It can be concluded that the majority of the study participants were found to have satisfactory knowledge, but the acceptance rate was comparatively lesser. Hence, more information and awareness campaigns must be launched reassuring the population about vaccine safety.

## Figures and Tables

**Table 1 vaccines-10-01605-t001:** Age group wise distribution of study participants based on their knowledge regarding COVID -19 Disease.

Knowledge (*n* = 606)															
Age Groups	Heard about COVID-19		Source of Knowledge (News)		Infected with COVID-19		Know the Causes of COVID-19 (Virus)		How Person Can Get COVID-19 (Direct Contact with Infected Person, Touching Infected Surface and Eating Contaminate Food)		Knows the Measures to Prevent COVID-19		Preferable Vaccine for COVID-19 (Covishield)		Total
	*n*	%	*n*	%	*n*	%	*n*	%	*n*	%	*n*	%	*n*	%	
16–25 years	166	97.6%	166	97.6%	46	27.1%	164	96.5%	42	24.7%	162	95.3%	88	51.8%	170
26–40 years	224	99.1%	208	92.0%	82	36.3%	202	89.4%	96	42.5%	196	86.7%	126	57.1%	226
41–60 years	162	100.0%	147	90.7%	62	38.38.3%	124	76.5%	62	38.3%	132	81.5%	84	51.9%	162
>60 years	48	100.0%	34	70.8%	8	16.7%	26	54.2%	22	45.8%	34	70.8%	16	33.3%	48

**Table 2 vaccines-10-01605-t002:** Gender wise distribution of study participants based on their knowledge regarding COVID-19 Disease.

Gender	Heard about COVID-19		Source of Knowledge (News)		Infected with COVID-19		Know the Causes of COVID-19 (Virus)		How Person Can Get COVID-19 (Direct Contact with Infected Person, Touching Infected Surface and Eating Contaminate Food)		Knows the Measures to Prevent COVID-19		Preferable Vaccine for COVID-19 (Covishield)		Total
	*n*	%	*n*	%	*n*	%	*n*	%	*n*	%	*n*	%	*n*	%	
Female	276	98.6%	252	90.0%	78	27.9%	236	84.3%	104	37.1%	242	86.4%	152	54.3%	170
Male	324	99.4%	303	92.9%	120	36.8%	280	85.9%	118	36.2%	282	86.5%	165	50.6%	226
*p* value	0.422		0.268		0.024		0.607		0.203		0.931		0.186		

**Table 3 vaccines-10-01605-t003:** Age group wise distribution of study participants based on their acceptance for COVID -19 vaccine.

Age Groups	Are You Vaccinated		In Future Which Vaccine You Would Prefer to Get Vaccinate (Covishield)		Even after Getting Vaccination, You Still Have Chances to Get COVID Disease		Know about the Side Effects of COVID-19 Vaccine (Fever, Headache, Pain, Swelling, Fatigue)		Are You Afraid All Because of Side Effects of COVID Vaccine		Any Comorbidity Stopping You from Getting COVID Vaccine (Diabetes)		Recommend COVID-19 Vaccine to Others		Do You Think COVID-19 Vaccine Is Safe in Pregnant and Lactating Women		Do You Think COVID-19 Vaccine Is Safe in Children below 18 Years	
	*n*	%	*n*	%	*n*	%	*n*	%	*n*	%	*n*	%	*n*	%	*n*	%	*n*	%
16–25 years	152	89.4%	86	50.6%	162	95.3%	25	14.7%	80	47.1%	0	0%	168	98.8%	120	70.6%	84	49.4%
26–40 years	190	84.1%	118	52.2%	218	96.5%	21	9.3%	136	60.2%	0	0%	218	96.5%	144	63.7%	106	46.9%
41–60 years	132	81.5%	66	40.7%	140	86.4%	20	12.3%	126	77.8%	4	2.5%	144	88.9%	106	65.4%	70	43.2%
>60 years	26	54.2%	18	37.5%	34	70.8%	8	16.7%	32	66.7%	2	4.2%	26	54.2%	20	41.7%	18	37.5%

**Table 4 vaccines-10-01605-t004:** Gender wise distribution of study participants based on their acceptance for COVID-19 vaccine.

Gender	Are You Vaccinated		In Future Which Vaccine You Would Prefer to Get Vaccinate (Covishield)		Even after Getting Vaccination, You Still Have Chances to Get COVID Disease		Are You Afraid All Because of Side Effects of COVID Vaccine		Any Comorbidity Stopping You from Getting COVID Vaccine (Diabetes)		Recommend COVID-19 Vaccine to Others		Do You Think COVID-19 Vaccine Is Safe in Pregnant and Lactating Women		Do You Think COVID-19 Vaccine Is Safe in Children below 18 Years	
	*n*	%	*n*	%	*n*	%	*n*	%	*n*	%	*n*	%	*n*	%	*n*	%
Female	228	81.4%	127	45.4%	248	88.6%	162	57.9%	6	2.1%	258	92.1%	168	60.0%	128	45.7%
Male	272	83.4%	161	49.4%	306	93.9%	212	65.0%	0	0.0%	298	91.7%	222	68.1%	150	46.0%
*p* value	0.522		0.291		0.028		0.078		0.010		0.769		0.041		1.000	

**Table 5 vaccines-10-01605-t005:** Overall Knowledge.

Ouestions Asked:	*n*	%
1 Have you ever heard about COVID-19 disease	600	99.0%
2 From where did you hear about COVID-19 disease (news)	555	91.6%
3 Have you ever been infected with the COVID disease	198	32.7%
4 According to you what are the cause of COVID-19 disease (virus)	516	85.1%
5 Do you know how a person can get COVID-19 disease (Direct contact with infected person, touching infected surface and eating contaminate food)	222	36.2%
6 Do you know the measures to prevent COVID-19 disease	524	86.5%
7 According to you which vaccine is best for prevention of COVID-19 (Covishield)	317	52.3%

**Table 6 vaccines-10-01605-t006:** Overall acceptance.

Ouestions Asked:	*n*	%
Are you vaccinated	500	82.5%
In future which vaccine you would prefer to get vaccinate (covishield)	288	47.5%
Even after getting vaccination, you still have chances to get COVID disease	554	91.4%
Know about the side effects of COVID-19 vaccine (fever, headache, pain, swelling, fatigue)	74	12.2%
Are you afraid all because of side effects of COVID vaccine	374	61.7%
Any comorbidity stopping you from getting COVID vaccine (diabetes)	6	1.0%
Recommend COVID-19 vaccine to others	556	91.7%
Do you think COVID-19 vaccine is safe in pregnant and lactating women	390	64.4%
Do you think COVID-19 vaccine is safe in children below 18 years	278	45.9%

## Data Availability

Available request.

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
