# Peer review of "Knowledge and Acceptance of the COVID-19 Vaccine for COVID-19 Disease Prevention among the Indian Population: A Mixed-Method Study"

_vaccines, 2022, doi:10.3390/vaccines10101605_

Round 1

Reviewer 1 Report

Comments –

The article entitled “Knowledge and Acceptance of COVID-19 vaccine for COVID-2 19 disease prevention among Indian population: A Mixed-3 Method Study” by Pratibha Taneja and colleagues present an interesting work by describing the two sets of information materials including “knowledge” and “acceptance’’ of COVID-19 disease and vaccination among the Indian population and by providing suggestions on how to counter the anti-vaccination campaign among the common majority of people of different ages and gender. However, there are few main flaws in the said work apart from typo and drafting’s errors as well as English language issues.

Major comments:

1.       The study is carried out quantitatively considering 606 participants via an E-survey while the qualitative part includes only 30 participants, I wonder how these two groups can be correlated statistically where there is a huge gap among them.

2.        Wouldn’t it be better to extrapolate the qualitative study by increasing the number of participants from different localities within the same district or among districts/provinces involving hospitals data? I am sure there would be enough data with these health care institutions which could be used in a broader way to cover the qualitative and quantitative opinions of a large number of participants.

3.       In the current format, this study falls very short by describing “among the Indian population” whereas only a limited number of data is presented.

4.       In my opinion, if such a study is carried out even among the affiliated universities where this work was carried out, I believe the study could be improved in terms of both qualitative and quantitative analyses.

5.       I would suggest that the authors should either restrict this study to only the locality where this study has been carried out and then do a statistical analyses considering the whole population (age, class, genders) and then correlate with other localities or they should consider all other data from other hospitals of a single district and then extend their statistical probabilities to other parts of the country thereby covering a nation-wide statistical analyses.

Author Response

Thank you so much for taking out time to review our paper and we would like to positively respond to all the comments. 

Reviewer 2 Report

The topic is certainly of scientific and popular interest, and can potentially bring an empirically-based case study forward to advance knowledge on the state of the art on this subject using India to illustrate trends and processes in the mass vaccination of populations against COVID-19. However, a number of problematic issues and matters arise from the execution and foundational assumptions of this study. 

1) The authors should avoid sweeping statements and generalities, such as endnote 6 on vaccinations saving millions of lives. This is simplified conjecture, given the very low infection fatality rate of the virus, and many used a case fatality rate measure that excluded co-morbidity factors. Also we can see a growing amount of evidence of huge numbers of injuries and deaths resulting from the vaccinations globally. 

2) I certainly would not characterise the vaccination acceptance or skepticism as being unknown, there is conjecture and preliminary results on this very issue. 

3) The virus is not deadly by its infection fatality rate, which is incredibly low. Furthermore, the current state of the art suggests that mass vaccinations are not in fact helping, and in regions or countries with a high level of vaccination also endure repeated waves of COVID-19! Many of the side effects dismissed in this submission have been confirmed in scientific reports by Pfizer and other drug companies. 

4) Rather than being an objective scientific inquiry, the submission reads as a form of activism for COVID-19 vaccinations that cherry picks supportive data for its conclusions. 

5) Rather than focusing upon constructed ethical evaluations of the sample based on whether they are or are not vaccinated, it would be far more scientifically productive if the focus was on the understanding of the genuine reasons for vaccine uptake or refusal without judgement. 

6) The conclusion is remarkably brief and very uninformative. It should answer the research question/hypothesis posed in the beginning of the work and to detail the main lessons learned and knowledge created. 

Author Response

Thank you for taking out time to review our paper. We offer all the responses to the comments one by one.

Round 2

Reviewer 2 Report

The revision is definitely an improvement on the original submission. However, some issues still need to be addressed. First of all the acknowledgement of the growing level of scientific evidence of the problems caused by the vaccinations, such as the Sudden Adult Death Syndrome or the recent admission by the head of Pfizer of the problems in the vaccine where he tried to blame governments for rushing the process of vaccinations. Furthermore, facts and figures tend to be given without context, yes over 5 million deaths are a large figure (not clear if they died with or of the virus hence CFR or IFR not known), but the Earth's population is currently over 7 billion! So there should be some more balance in presentations of the cons as well as the pros of vaccination and this should help to better understand the responses of the respondents. 

Author Response

Review response

We would like to explicitly thank the reviewer for taking out their time to help review this paper and provide suggestions to improve the paper. We would like to provide our response below

Reviewer comment:

The revision is definitely an improvement on the original submission. However, some issues still need to be addressed. First of all the acknowledgement of the growing level of scientific evidence of the problems caused by the vaccinations, such as the Sudden Adult Death Syndrome or the recent admission by the head of Pfizer of the problems in the vaccine where he tried to blame governments for rushing the process of vaccinations. Furthermore, facts and figures tend to be given without context, yes over 5 million deaths are a large figure (not clear if they died with or of the virus hence CFR or IFR not known), but the Earth's population is currently over 7 billion! So there should be some more balance in presentations of the cons as well as the pros of vaccination and this should help to better understand the responses of the respondents. 

Author response: As advised by the reviewer we have added data related to the Sudden Adult Death syndrome and also talked about the cons or limitations of taking the vaccine including putting the patients at higher fatality risk.